# Insights into sea surface temperature variability and the impact of long-term warming on marine heatwaves in the Mediterranean Sea

Dimitra Denaxa[1,2], Gerasimos Korres[1], Sofia Darmaraki[3], Maria Hatzaki[2]

[1] Hellenic Centre for Marine Research (HCMR), Institute of Oceanography, Anavyssos, Greece
[2] National and Kapodistrian University of Athens, Department of Geology and Geoenvironment, Athens, Greece
[3] Coastal & Marine Research Laboratory (CMRL), Institute of Applied and Computational Mathematics (IACM), Foundation for Research and Technology—Hellas (FORTH), Crete, Greece

*Correspondence to*: Dimitra Denaxa (ddenaxa@hcmr.gr)

**Abstract.** In the context of a warming Mediterranean Sea, marine heatwaves (MHW) have progressively intensified, leading to multiple environmental and socioeconomic damage. This study explores the origin of the observed trends in surface MHWs in the basin using sea surface temperature (SST) observations for the period 1982–2023. Results show a basin-wide increase of SST and extreme SST occurrences over the study period, emphasized in the eastern basin. The Adriatic, Aegean and northern Levantine Seas exhibit the highest trends of SST as well as extreme SST percentiles, suggesting that these are

the most vulnerable areas in the basin both in terms of accumulated warming and extreme SST occurrences. On top of the underlying mean warming, increased variability of SST is observed in parts of the western and central Mediterranean Sea, while decreased variability of SST is found in most of the eastern basin. Results reveal a basin-wide dominance of mean warming versus interannual variability in causing higher maximum MHW intensities, more extreme MHWs, longer heat exposure as well as a greater accumulation of heat stress on an annual basis. Interannual variability becomes the dominant

driver of the mean MHW intensity trends in most of the basin and particularly in western and central Mediterranean areas. Mean MHW intensity is differentiated from the other examined metrics also due to the higher sensitivity of our trend attribution results for this metric to different methodological choices for climatological baselines, thus implying a more complex nature of this metric. To advance our understanding of forcing factors behind MHW trends in the Mediterranean Sea, future work should incorporate climate models, which can explicitly represent the anthropogenic nature of trends

against natural ocean variability.

## 1. Introduction

Over the past four decades, the Mediterranean Sea has undergone continuous warming of the sea surface temperature (SST) at observed rates ranging from 0.37 to 0.41 °C per decade between 1982–2022 (e.g., Mohamed et al., 2019; Pisano et al., 2020; Pastor et al., 2020; Juza et al., 2021; EU Copernicus Marine Service Product, 2022, Martinez et al., 2023).

Additionally, extreme warm events, named as marine heatwaves (MHWs), have garnered significant attention due to their

severe environmental and socioeconomic impacts and their notable intensification over recent decades (Darmaraki et al., 2019a; Juza et al., 2022; Dayan et al., 2023; Hamdeno and Azcarate, 2023, Pastor and Khodayar, 2023; Denaxa et al., 2024).

Recent studies have addressed the origin of observed MHW trends in the Mediterranean Sea by assessing the relative contributions of long-term changes in mean SST and changes in SST variability. Simon et al. (2023) investigated the role of the long-term SST warming (mean warming from now on) and interannual variability of SST in altering summer MHW activity. They showed that the mean warming is the main driver of the increase in MHW activity across the basin. They also showed that the SST variability contributes to long-term trends of MHW activity in the western and Adriatic regions while it acts towards reducing these trends in central and eastern Mediterranean regions. Ciappa (2022) and Martinez et al. (2023) also showed that MHW intensification mainly results from the mean warming in the basin. Moreover, Oliver et al. (2019) used a statistical climate model to simulate trends of surface MHWs originating from mean warming or changes in variance of SST in the global ocean. Their results for the Mediterranean Sea show that trends in the annual number of MHW days and the maximum MHW intensity are mainly driven by the mean warming. The latter has been also suggested as the primary driver behind observed trends in properties of surface MHWs at coastal locations globally by Marin et al. (2021). In particular, they quantified the mean warming contribution versus that of interannual variability using a trend attributional ratio based on an ensemble approach. Their results reveal that mean warming is mainly responsible for the observed trends in all examined MHW metrics except for mean MHW intensity, whose trends are largely attributed to internal variability.

As a biodiversity hotspot and one of the most sensitive marine regions to climate change, the Mediterranean Sea necessitates a comprehensive, basin-wide investigation of MHWs, accounting for areas that support key ecological processes susceptible to disruptions by MHWs. For instance, pelagic species may be affected by MHWs, with potential repercussions for the fishery industry. A climate risk assessment by Hidalgo et al. (2022) identifies the southeastern basin as the most impacted for both pelagic and demersal fisheries, highlighting geographic differences in drivers and impacts and recommending regionally tailored adaptation strategies. In addition, MHWs pose significant risks to aquaculture, a rapidly expanding industry in the Mediterranean Sea, by increasing fish mortality and facilitating the proliferation of pathogens and disease outbreaks, which can lead to substantial financial losses (Cascarano et al., 2021; Mengual et al., 2021). A comprehensive understanding of MHWs across the basin can further inform conservation measures in Marine Protected Areas (MPAs), which host vulnerable species, such as marine mammals and turtles (Chatzimentor et al., 2023). In fact, a regional-scale approach can provide critical insights for identifying spatial refugia and establishing new MPAs, strengthening the resilience of Mediterranean marine ecosystems to climate change (e.g., Zentner et al., 2023; Bates et al., 2019).

Given the broad ecological and socio-economic challenges associated with Mediterranean MHWs, this study investigates extreme warming conditions and the effect of mean SST warming versus that of interannual variability of SST on MHW trends across the basin. We first provide insights into changes in the Mediterranean SST over the 42-year period using

observational SST data. This part provides essential context on long-term changes in extremes and variability throughout the basin, setting the stage for the subsequent MHW analysis while also highlighting areas where extreme warming is particularly pronounced and might be disproportionately impacted. Then, we quantify the relative contributions of mean SST warming and interannual variability of SST to observed trends of the selected MHW properties, applying the methodological approach proposed by Marin et al. (2021) to the entire Mediterranean basin.

**2. Data and Methods**

**2.1 SST observations**

For the SST analysis and detection of MHWs in this study we use satellite SST observational data in the Mediterranean Sea for the period Jan 1982 – Dec 2023. Gridded SST data at daily frequency and 0.05 ° x 005 ° horizontal resolution are obtained from the Reprocessed satellite SST dataset of the Copernicus Marine catalogue for the study period (product ref.
no. 01 (Table 1)). This reprocessed product provides optimally interpolated estimates of the foundation SST based on satellite observations across the Mediterranean Sea and is extensively validated against drifter buoy measurements as outlined in its Quality Information Document (Pisano et al., 2023), confirming its suitability for MHW studies. For computational reasons, the SST values are re-gridded to an 0.25 ° x 025 ° spatial grid.

De-seasonalised daily SST anomaly time-series are constructed by subtracting the daily climatological SST for the period
1982–2023 at each grid point, thus removing the effect of seasonal fluctuations. To investigate changes on extreme SST occurrences, the 99[th] percentile (P99) of the daily SST anomalies of each year and the corresponding trend over the study period are also computed. In addition, the standard deviation (STD) of the daily SST anomalies is calculated over each year to obtain an annual measure of SST variability. The linear trend of the annual STD time series is computed to gain insights into changes in SST variability over the study period. Trends are computed using linear regression, and their statistical
significance is assessed using the Mann-Kendall test at the 95% confidence level. Confidence intervals for the trend estimates are calculated based on the standard error of the regression coefficients, using the Student's t-distribution ($\alpha=0.05$).

To exclude long-term variations, we subsequently remove the linear trend over the 42-year study period from the SST anomaly time series, creating a detrended daily SST dataset. This additional dataset is used to investigate MHWs independently of the long-term warming in the basin. Notably, we assume that the variability of SST in the detrended dataset
may arise from internal ocean variability as well as from potential indirect effects of the climate-change signal, given the ocean inertia and the complex ocean dynamics at multiple temporal and spatial scales. On these grounds, we use the term interannual –rather than internal– variability, to address the remaining temperature variations after removing the long-term trend component.

**Table 1**

| Product ref. no | Product ID & type | Data Access | Documentation |
|---|---|---|---|
| 1 | SST_MED_SST_L4_REP_OBSERVATIONS_010_021; Satellite observations<br><br>Mediterranean Sea High Resolution L4 Sea Surface Temperature Reprocessed | EU Copernicus Marine Service Product (2023) | Product User Manual (PUM): Pisano et al. (2023)<br><br>QUality Information Document (QUID): Pisano et al. (2023)<br><br>https://catalogue.marine.copernicus.eu/documents/QUID/CMEMS-SST-QUID-010-021-022-041-042.pdf<br><br>https://catalogue.marine.copernicus.eu/documents/PUM/CMEMS-SST-PUM-010-021-022-041-042.pdf |


## 2.2 MHW analysis: detection and origin of trends

MHWs are defined upon the exceedance of a locally determined climatological threshold (90[th] percentile) of SST for a 5-day period at each grid point, following Hobday et al. (2016) and are identified using the MATLAB toolbox by Zhao and Marin (2019).

To investigate the effect of mean warming and interannual variability on MHW trends, we follow Marin et al. (2021), where the origins of observed trends in MHW metrics are assessed using a trend attributional ratio (TAR). We first identify MHWs on the daily SST anomaly timeseries between 1982–2023, where long-term SST trends are included (named as non-detrended dataset from now on). Subsequently we detect MHWs that emerge solely due to interannual variability within the study period, by applying the MHW detection algorithm on the detrended dataset. We then use the two MHW detection 105 outputs to compute selected metrics, listed in Sect. 2.3. To isolate the effect of the mean SST warming on a particular MHW metric, we remove the interannual variability (iv) component from the observed (obs) value of that metric, as follows:

$$M = M_{obs} - M_{iv} \qquad (Eq.\ 1)$$

where $M_{obs}$ and $M_{iv}$ are the annual values of the metric obtained from the non-detrended and the detrended dataset respectively. Linear trends of $M$ and $M_{iv}$ are then computed for the entire Mediterranean Sea, as well as for the western, 110 central and eastern sub-basins, which are separated by the strait of Sicily and a fixed-longitude boundary at 22° E,

respectively (Fig. 1a). Following Marin et al. (2021), the TAR is obtained at each grid point by computing the linear trend of M and $M_{iv}$ component for each MHW metric, according to the following equation:

$$TAR = \left(|Trend_M| - |Trend_{M_{iv}}|\right)/\max\left(|Trend_M|, |Trend_{M_{iv}}|\right) \quad \text{(Eq. 2)}$$

where $Trend_M$ and $Trend_{Miv}$ are the trends of the M and $M_{iv}$ components representing MHW properties due to long-term SST changes and interannual variability, respectively. Scaling this difference with the maximum absolute value of the aforementioned trends forces TAR to range from -1 to 1. Positive (negative) TAR values indicate a stronger role of the mean warming (interannual variability) in forming the observed trends of a given MHW property.

A daily SST climatology of the detrended SST dataset is constructed following Hobday et al. (2016) based on the entire period (1982–2023). Similar to Marin et al. (2021), this climatology is used for the detection of MHWs in both the non-detrended and detrended datasets. MHWs derived from both datasets are therefore relative to the initial state of the study period (1982), offering insights into what would have occurred without the mean warming effect. This approach for the climatological baseline aims to better represent the impact of long-term changes in MHW metrics and facilitates a direct inter-comparison with the findings of Marin et al. (2021) for Mediterranean coastal locations. Nonetheless, to examine the sensitivity of TAR values of the examined metrics to the approach followed for the climatological baseline, two extra experiments are carried out (EXP2 and EXP3) in addition to the initial approach (EXP1). In EXP2, the aforementioned methodology for computing TAR is repeated after detecting MHWs in the non-detrended dataset based on its own climatology (1982–2023), instead of using the climatology obtained from the detrended dataset as in EXP1. EXP3 follows the climatology computation approach of EXP2 but reduces the reference period to its first half (1982–2002) to explore the impact of using a shorter reference period on the attribution of trends.

## 2.3 Selected MHW metrics

To complement existing knowledge and provide impact-related insights, we examine the yearly cumulative intensity and the normalized maximum event intensity, in addition to the mean MHW intensity (SST anomaly with respect to climatology averaged over the event duration), maximum MHW intensity (maximum SST anomaly with respect to climatology over the event duration) and the total number of MHW days (total count of days with a MHW activated).

The yearly cumulative intensity ($CI_{yearly}$) serves as a measure of the long-term thermal stress induced by MHWs, combining the effect of mean intensity ($I_{mean}$) and duration accumulated on a yearly basis. Considering the total events (N) of the year, it is computed as follows:

$$CI_{yearly} = \sum_{i=1}^{N} I_{mean} \cdot duration \quad \text{(Eq. 3)}$$

The normalized maximum event intensity is indicative of the degree to which SST exceeds the local climatology. It is defined as the peak intensity over the event (i.e., $SST_{max} - SST_{clim}$) divided by the deviation of the detection threshold ($SST_{P90}$) from the corresponding daily climatology ($SST_{clim}$), as in Gupta et al. (2020):

$$SI = (SST_{max} - SST_{clim})/(SST_{P90} - SST_{clim}) \quad \text{(Eq. 4)}$$

Values ranging from (1–2], (2–3], (3–4] and above 4 correspond to moderate, strong, severe and extreme conditions, respectively (Hobday et al., 2018). Here, we choose to focus on the maximum value of this index ($SI_{max}$) each year. The $SI_{max}$ represents MHW severity by means of capturing the most extreme temperatures with respect to local SST variability, regardless of the event duration, thus informing on the "local worst" extremely warm instances occurring on a yearly basis.

As different marine organisms exhibit different i) upper thermal limits, ii) adaptation capacity to local temperature variations and iii) ability to geographically shift in order to avoid excess heat stress, usefulness of $CI_{yearly}$ and $SI_{max}$ relies on the context of specific impact assessment studies and application. For instance, the cumulative effect of multiple MHWs (represented by $CI_{yearly}$) may be relevant for several species, such as gorgonian populations which have been severely impacted by recurrent events in the basin (Orenes-Salazar et al., 2023) or fish species, such as the gilthead seabream, whose thermal tolerance can be affected by past exposure to thermal stress (Kir, 2020). Such species may also be vulnerable to extreme temperature peaks (represented by $SI_{max}$). For example, unlike wild marine species that can often escape unfavorable aquatic conditions, farmed species confined to aquaculture environments are more vulnerable to warm extremes (Beever et al. 2017). Such distinctions in behavioral adaptability are important for selecting appropriate metrics to assess MHW impacts on different marine populations and environments.

## 3. Results

### 3.1 Mean and extreme SST trends

This section examines the trends and spatial distribution of the observed mean (Fig. 1a) and extreme (Fig.1b) warming of SST in the Mediterranean Sea for the period 1982–2023. Results reveal a continuous surface warming over the past 42 years both in terms of mean annual and extreme warming conditions, in agreement with previous studies (e.g., Pisano et al., 2020, Pastor et al., 2020). In our study, the average warming trend for the basin for the examined period is 0.41 degC.dec[-1]. The warming trend is statistically significant across the basin, with higher values observed over the eastern Mediterranean Sea, reaching up to 0.6 degC.dec[-1] in the Aegean and northern Levantine Seas. The lowest warming trend is observed in the Alboran Sea (0.25 degC.dec[-1]), followed by the Gulf of Lions and the southwestern Ionian Sea (0.3 degC.dec[-1]) (Fig. 1a).

Long-term changes on extreme SST occurrences are shown through the trend of the 99[th] percentile of SST anomalies (P99) depicted in Fig. 1b. The P99 shows statistically significant positive trends across the entire basin, suggesting a basin-wide

intensification of extremely warm surface conditions (Fig. 1b). The highest trend values of P99 are observed in the Ligurian, Adriatic, northeastern Ionian and Aegean Seas, at about 0.6 degC.dec$^{-1}$ and locally exceeding 0.7 degC.dec$^{-1}$ (e.g., to the

southeast of Rhodes Island). Weaker warming trends of the P99 are observed in the western basin, the western Ionian and certain areas in the southern Levantine Sea, with minimum values in the Gulf of Lions (lower than 0.3 degC.dec$^{-1}$).

While long-term trends of SST anomalies (Fig. 1a) provide insights into the overall warming of the sea surface, they do not capture how temperatures fluctuate around the observed trend over the study period. In this respect, the linear trend of the STD of the daily SST anomalies informs on changes in their variability (Fig. 1c). Specifically, we observe statistically

significant increase of the STD mainly in areas of the western and central Mediterranean Sea. Maximum STD trend values are observed in the northwestern basin (eastern part of Gulf of Lions, Ligurian Sea) and the northern Adriatic Sea, indicating heightened variability of SST. Conversely, the area south of the Balearic Islands as well a great portion of the eastern Mediterranean basin exhibit statistically significant negative STD trend, indicating a weaker variability of SST towards the most recent years in these areas.


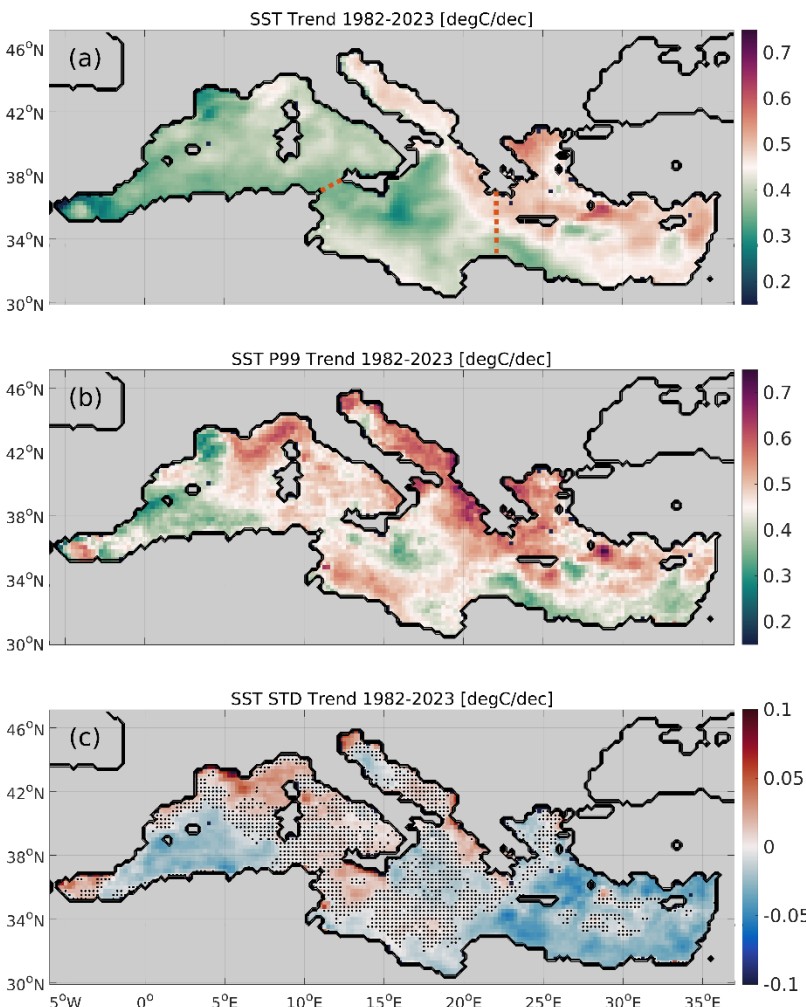

**Figure 1: Linear trend of (a) SST, (b) the 99th percentile (P99) of SST and (c) the standard deviation (STD) of SST, for the period 1982-2023, using de-seasonalized SST anomalies. Black dots superimposed on trend fields denote the locations with statistically insignificant trends (Mann Kendall test, 95% confidence level). The western, central and eastern sub-basins are separated by the strait of Sicily, and a fixed-longitude boundary at 22° E, respectively (red dashed lines in (a)). Product used: Mediterranean Sea High Resolution L4 Sea Surface Temperature Reprocessed (Table 1, product ref. no.1).**

Notably, the STD trend is independent of the mean warming trend and reflects changes in the variability of SST around mean annual values. Our findings therefore indicate that, on top of the underlying mean warming, few distinct areas, primarily in the western basin, experience increased variability, while most of the eastern basin and the region south of the Balearic Islands show reduced variability. These results are in agreement with Martinez et al. (2023) that compared the STD of SST anomalies between the periods 1982–2002 and 2002–2022 for different Mediterranean sub-basins, reporting a

decreased variability in the Ionian, Aegean and Levantine Seas and increased variability in the western basin, Adriatic and Tyrrhenian Seas during the most recent period.

## 3.2 Attribution of trends in MHW metrics

This section studies the relative role of the mean SST warming and the SST interannual variability on MHW trends. The temporal evolution and long-term trends of the selected MHW metrics (Sect. 2.3) derived from the non-detrended and the detrended dataset are presented in Fig. 2 for the entire Mediterranean Sea and its sub-basins.

Results show an increase in all MHW metrics derived from the non-detrended SST dataset over the study period (Fig. 3a-e, Table 2). Mean intensity of MHWs increases by 0.03 deg.dec$^{-1}$, with higher values in the western compared to the central
and eastern sub-basins (Fig. 2a, Table 2). The total number of MHW days shows significant positive trends, increasing from the western to the eastern basin, with approximately 16, 18, and 23 days per decade for the western, central, and eastern sub-basins, respectively (Fig. 2c). The basin-average, yearly cumulative intensity increases by ~33 degC.days.dec$^{-1}$, with its highest values and linear trends seen in the eastern sub-basin (Fig. 2d, Table 2).

The maximum intensity ($I_{max}$) shows a higher positive trend in the western (0.38 degC.dec$^{-1}$) than in the eastern sub-basin
(0.32 degC.dec$^{-1}$) (Fig. 2b). In contrast, the maximum severity index ($SI_{max}$) exhibits a stronger increasing trend in the eastern (0.3 units.dec$^{-1}$) than in the western sub-basin (0.23 units.dec$^{-1}$) (Table 2). This pattern arises potentially due to $SI_{max}$ quantifying the extremity of MHW intensity relative to a fixed climatological threshold (Eq. 4). Since this threshold remains constant over time, the larger increase in $SI_{max}$ in the eastern sub-basin suggests that maximum intensities in this region are becoming proportionally more extreme compared to historical conditions. The observed differences in trends between $I_{max}$
and $SI_{max}$ across sub-basins highlight that absolute MHW intensities are increasing more rapidly in the western sub-basin, while their relative extremity compared to the historical baseline is increasing more in the eastern sub-basin.

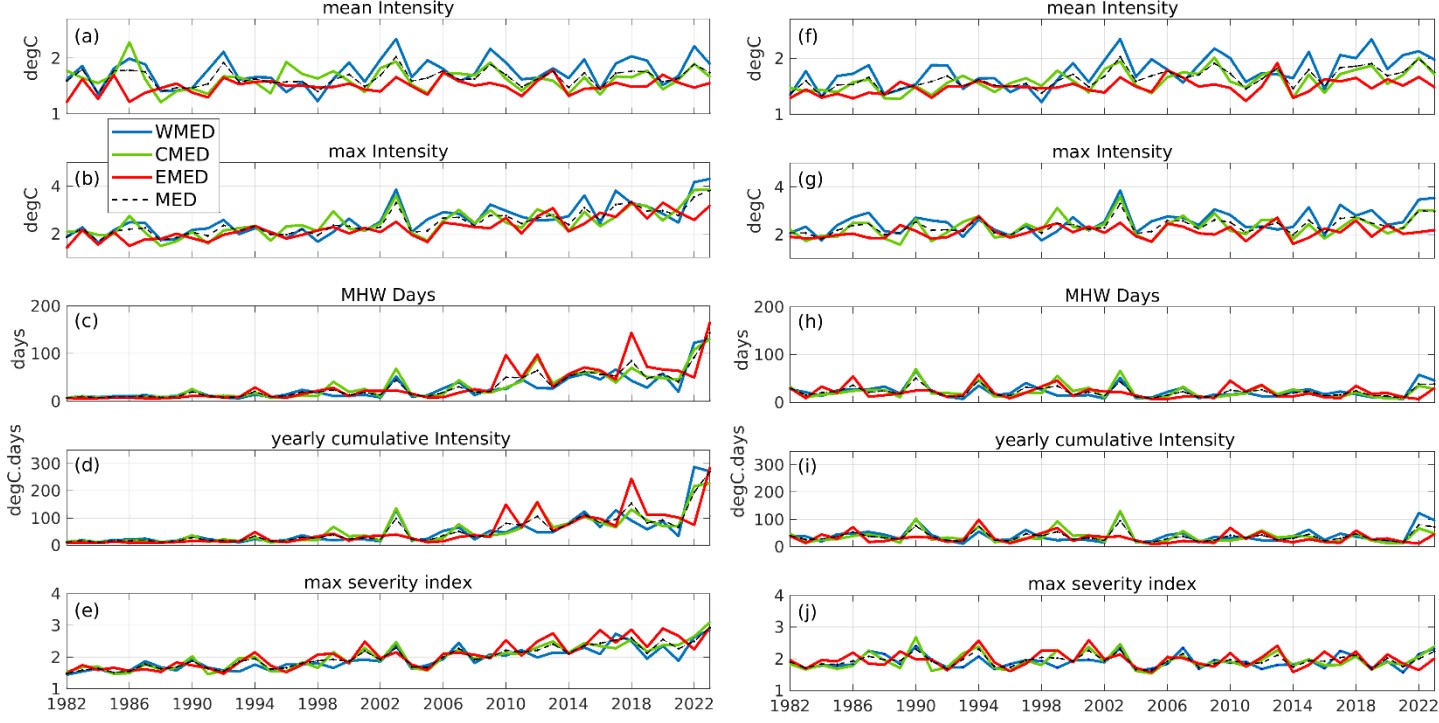

**Figure 2: Temporal evolution of MHW characteristics over 1982-2023 using the non-detrended (a-e) and the detrended SST anomaly timeseries (f-j). From top to bottom: Mean intensity (a,f), maximum intensity (g,b), Number of MHW days (c,h), yearly cumulative intensity (d,i), and maximum severity index (e,j). Blue, green and red lines and the black dashed line correspond to spatial averages for the western (WMED), central (CMED), eastern (EMED) Mediterranean sub-basins and the entire Mediterranean basin (MED), respectively. Product used: Mediterranean Sea High Resolution L4 Sea Surface Temperature Reprocessed (Table 1, product ref. no.1).**

MHW metrics originating from the detrended dataset represent MHW conditions that would have occurred solely due to interannual variability. Most metrics from this dataset show insignificant trends over the study period, with MHW intensity being a noteworthy exception (Fig. 2f-j, Table 2). In particular, both mean and maximum intensity exhibit higher trends in the western basin, similar to the non-detrended dataset. The statistically significant basin-wide positive trend of mean intensity ranges from 0.12 to 0.08, and 0.05 degC.dec$^{-1}$ for the western, central and eastern sub-basins, respectively (Fig. 2f). Interestingly, these trends are higher than those from the non-detrended dataset, suggesting that interannual variability alone would still result in a statistically significant increase in mean MHW intensity (Fig. 2a,f).

Also $I_{max}$ in the detrended dataset shows positive trends, though they are not significant for the central and eastern sub-basins (Fig. 2g). Despite this, $SI_{max}$ in the detrended dataset shows non-significant trends (Fig. 2j). In conjunction with the positive

$SI_{max}$ trends in the non-detrended dataset, this indicates that the observed increase in $SI_{max}$ is primarily driven by the long-term warming trend, particularly in the eastern sub-basin (Table 2).

Moreover, the annual number of MHW days derived from the detrended dataset shows an insignificant trend of -1.4 days.dec$^{-1}$ for the entire basin, with a larger decrease for the eastern sub-basin (Fig. 2h, Table 2). Similarly. the yearly cumulative intensity derived from the detrended dataset shows non-significant trends (Fig. 2i), with higher (absolute) values again found in the eastern sub-basin (-2.28 degC.days.dec$^{-1}$). This aligns with the trends in MHW days, which constitute an annual measure of the total duration of MHW conditions and are therefore closely linked to the yearly cumulative intensity.

These results potentially suggest that the decreasing duration of MHW conditions in the detrended dataset (Fig. 2h) counterbalance the rising mean intensity (Fig. 2f) leading to a lower accumulation of heat stress as represented by yearly cumulative intensity (see Eq. 3).

Overall, with the exception of MHW intensity, our results suggest that interannual variability mainly tends to dampen the climate-change driven increase in MHW characteristics in the Mediterranean Sea, particularly in the eastern sub-basin, in
agreement with Simon et al. (2023). This observation is in line with results presented in Sect. 3.1 showing an increase (decrease) in SST variability mainly in western (eastern) Mediterranean areas throughout the study period. Nevertheless, further investigations are needed to unravel the cause of the differentiation of MHW intensity compared to other MHW properties.

**Table 2: Linear trends and confidence intervals for the spatially averaged MHW metrics of the non-detrended (N-DET) and the detrended (DET) SST dataset over 1982–2023. Metrics used: Mean intensity ($I_{mean}$), maximum intensity ($I_{max}$), Number of MHW days (MHW days), yearly cumulative intensity ($CI_{yearly}$), and maximum severity index ($SI_{max}$). Sub-regions considered: western (WMED), central (CMED), eastern (EMED) Mediterranean and the entire Mediterranean basin (MED). Values in bold correspond to statistically significant trends (Mann Kendall test, 95% confidence level). Product used: Mediterranean Sea High**
**Resolution L4 Sea Surface Temperature Reprocessed (Table 1, product ref. no.1).**

| | | MED | | WMED | | CMED | | EMED | |
|---|---|---|---|---|---|---|---|---|---|
| | | Trend | CI (+/-) | Trend | CI (+/-) | Trend | CI (+/-) | Trend | CI (+/-) |
| $I_{mean}$ **trend (degC/dec)** | N-DET | 0.03 | 0.04 | 0.06 | 0.06 | 0.01 | 0.05 | 0.03 | 0.03 |
| | DET | **0.09** | 0.04 | **0.12** | 0.06 | **0.08** | 0.04 | **0.05** | 0.03 |
| $I_{max}$ **trend (degC/dec)** | N-DET | **0.34** | 0.08 | **0.38** | 0.120 | **0.32** | 0.11 | **0.32** | 0.07 |
| | DET | **0.11** | 0.07 | **0.17** | 0.11 | 0.11 | 0.11 | 0.04 | 0.07 |
| **MHW days trend** | N-DET | **18.41** | 4.62 | **15.69** | 5.21 | **18.22** | 5.02 | **22.71** | 6.56 |

| | | | | | | | | | |
|---|---|---|---|---|---|---|---|---|---|
| (days/dec) | DET | -1.40 | 2.68 | -0.94 | 3.55 | -1.95 | 3.75 | -2.11 | 3.27 |
| $CI_{yearly}$ trend (degC.days/dec) | N-DET | **33.20** | 9.31 | **32.20** | 12.07 | **31.47** | 9.6 | **36.75** | 11.25 |
| | DET | 0.13 | 4.94 | 1.90 | 7.12 | -1.23 | 6.60 | -2.28 | 5.01 |
| $SI_{max}$ trend (units/dec) | N-DET | **0.26** | 0.05 | **0.23** | 0.06 | **0.26** | 0.06 | **0.30** | 0.06 |
| | DET | 0.01 | 0.05 | 0.01 | 0.06 | 0.02 | 0.07 | -0.01 | 0.07 |

To attribute MHW trends over the study period to the mean warming or the interannual variability of SST, TAR values are computed for the examined MHW properties (Fig. 3). As explained in Sect. 2.2, positive TAR values correspond to a major role of the mean SST warming while negative values correspond to a major role of interannual variability in driving MHW trends.

Results reveal a basin-wide dominance of the mean warming in driving the observed trends of all MHW characteristics, except for the mean MHW intensity that is found to be mainly forced by interannual variability over most of the Mediterranean Sea (Fig. 3a). In particular, a great part of the western and most of the central basin reaching up to 26° E exhibit large negative TAR values, often reaching -1, indicating that the mean MHW intensity trend is primarily driven by interannual variability. In contrast, the mean SST warming appears to dominate interannual variability over the eastern Levantine and part of the Aegean Seas, as well as in certain areas in the southwestern basin, as indicated by high positive mean intensity TAR values in these areas (Fig. 3a). These positive TAR locations largely coincide with the ones showing a decreasing STD of SST anomalies over the study period (Fig. 1c, Fig. 3a). The dominance of the mean warming signal in explaining the MHW intensity trends in these areas appears associated with the decreasing trend of the variability of SST in the same areas. We note that the positive mean intensity TAR values in these areas result from significant positive trends of the mean warming component and insignificant trends of the interannual variability component for this metric (not shown).

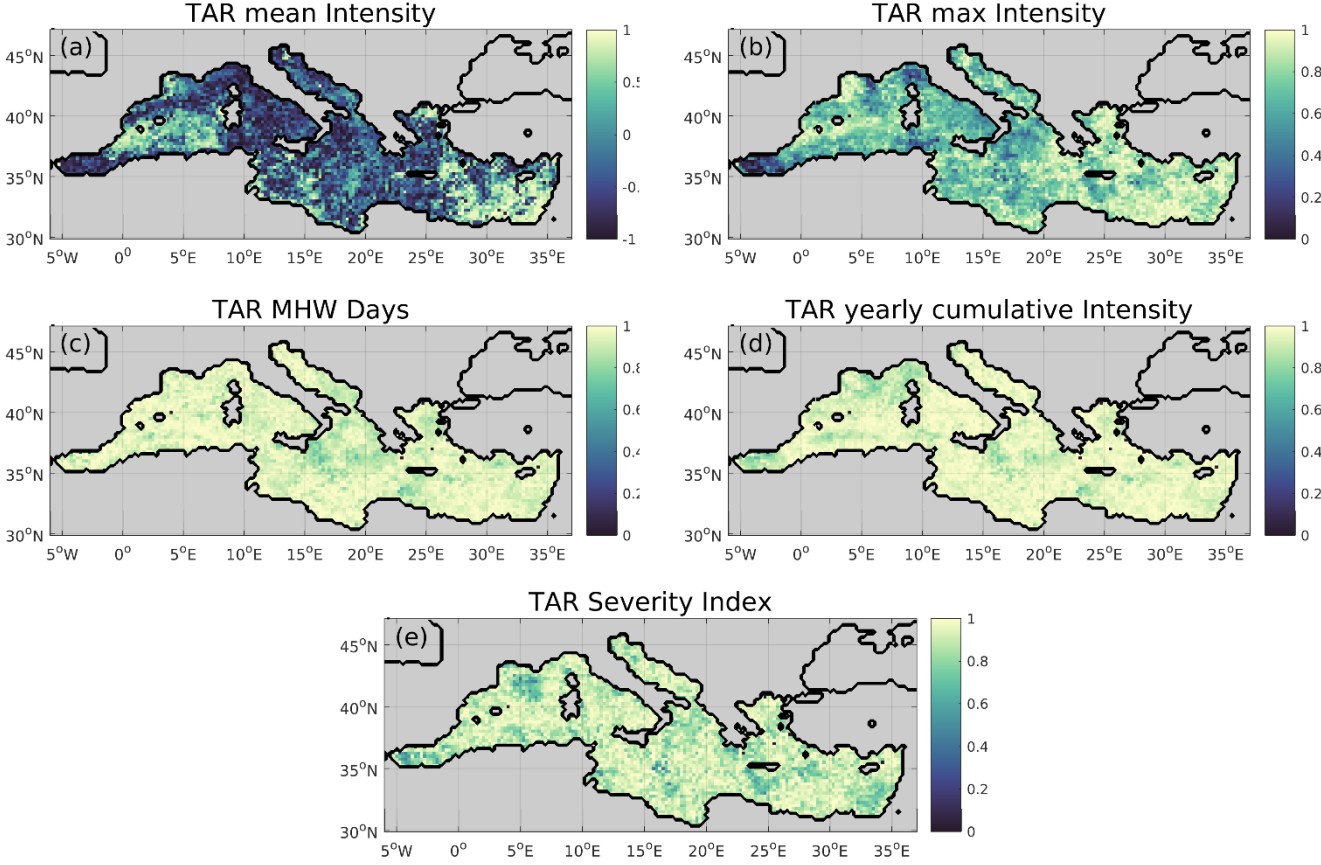

**Figure 3: Trend Attributional Ratio (TAR) for MHW characteristics over 1982-2023: (a) mean MHW intensity, (b) maximum MHW intensity, (c) number of MHW days, (d) yearly cumulative MHW intensity and (e) maximum MHW severity index. Positive (negative) TAR values indicate that mean SST warming (interannual variability) is the dominant driver of the trend in a MHW metric. Range of TAR values in colorbar axes vary for mean intensity and the rest of the variables to allow for visualizing spatial variations. Product used: Mediterranean Sea High Resolution L4 Sea Surface Temperature Reprocessed (Table 1, product ref. no.1).**

For the rest of the examined MHW properties' trends, positive TAR values across the entire basin indicate a dominant role of mean SST warming in driving their increase during 1982-2023 (Fig. 3b-e). We note here that the greater the value of TAR, the stronger the impact of the mean SST warming is. Positive TAR values for the maximum MHW intensity shown in Fig. 3b range from 0 to 1 across the basin, revealing areas with a marginally stronger influence of the mean SST warming signal compared to that of the interannual SST variability (Alboran, followed by the Ligurian Sea). However, there seems to be a clear prevalence of the mean SST warming especially over the eastern basin for this metric (Fig. 3b). Spatial distribution

of TAR for the maximum severity index exceeds 0.6 in most of the basin, indicating a basin-wide dependence of this metric on the mean SST warming (Fig. 3e). Furthermore, high positive TAR values are seen for the number of MHW days and the yearly cumulative intensity, exceeding 0.8 throughout the basin (Fig. 3c,d).

Overall, from a basin-wide perspective, our findings suggest a dominant role of the mean SST warming in the development of more severe and prolonged MHW conditions. By contrast, interannual variability becomes the dominant driver of the mean MHW intensity trend in most of the basin, playing an important role as well on the maximum MHW intensity trends in certain Mediterranean areas, though to a notably lesser extent.

To examine the sensitivity of TAR values of the examined metrics to the approach followed for climatological baseline, two
additional tests are performed as described in Sect. 2.2. These experiments reveal minimal TAR sensitivity for all metrics except for mean intensity (Fig. 4). The bar graphs of Fig. 4 summarize this finding showing the number of locations (as a percentage of the total number of Mediterranean grid points) exhibiting a positive TAR sign (i.e., a dominant role of the mean warming in MHW trends). In all experiments, the long-term trends of maximum intensity, yearly cumulative intensity, maximum severity index and annual number of MHW days are mainly attributed to the mean warming across more than
97% of the Mediterranean Sea. On the other hand, we observe a differentiated sensitivity of the mean intensity TAR to the examined approaches for climatology. Based on the initial approach (EXP1), 34% of the Mediterranean Sea shows mean warming-driven mean intensity trends, while for the 2$^{nd}$ and 3$^{rd}$ approach this percentage is 46% (EXP2) and 30% (EXP3) (Fig. 4). We note that the spatial distribution of mean intensity TAR values is similar in the three experiments (the same applies for the rest of the metrics) with the easternmost and a part of the southwestern basin showing a dominant role of
mean warming (not shown), as appears in Fig. 3a. Notably, mean MHW intensity is the metric found to be largely associated with interannual variability, as opposed to the rest of the examined MHW properties. The higher sensitivity of our results for mean intensity to different methodological choices further suggests a less predictable future behavior for this MHW property.

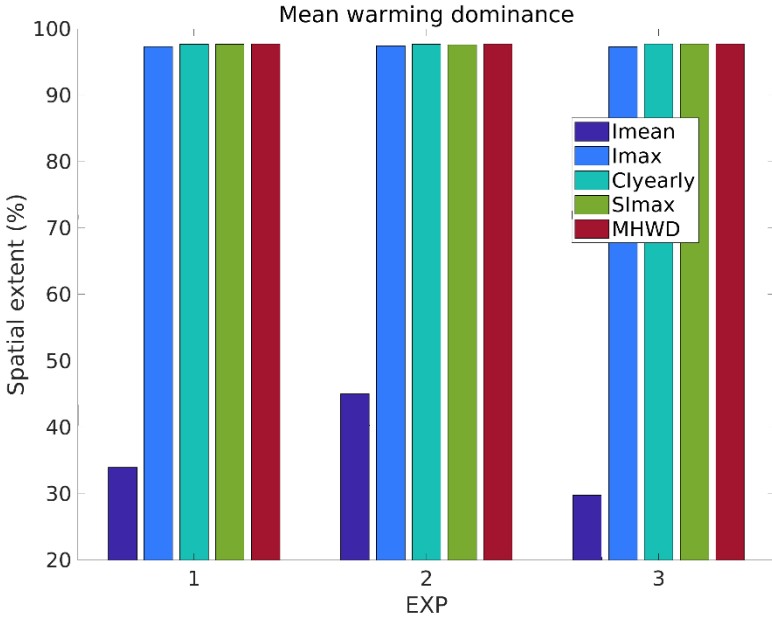

**Figure 4: Sensitivity of the Trend Attributional Ratio (TAR) to different approaches for base climatology used in MHW detection in the original (non-detrended) dataset. Vertical axis: Percentage of Mediterranean locations with positive TAR (i.e., mean warming dominance in trends of MHW metrics). Horizontal axis: (EXP 1) Climatology computed based on the detrended dataset with reference period 1982–2023; (EXP 2) Climatology computed based on the non-detrended dataset with reference period 1982– 2023; (EXP 3) as EXP 2 but with reference period 1982–2002. Metrics used: Mean intensity (Imean), maximum intensity (Imax), Number of MHW days (MHWD), yearly cumulative intensity (CIyearly), and maximum severity index (SImax). Product used: Mediterranean Sea High Resolution L4 Sea Surface Temperature Reprocessed (Table 1, product ref. no.1).**

## 4. Summary and conclusions

This study investigates the role of long-term changes in SST on observed trends of MHWs in the Mediterranean Sea, using SST observations for the period 1982–2023. Our results show a basin-wide increase in SST and extreme SST occurrences over the study period, with stronger trends in the eastern basin. The Adriatic, Aegean and northern Levantine Seas show the highest trends of both SST and its 99[th] percentile, highlighting these areas as particularly vulnerable both in terms of accumulated warming and extreme SST occurrences. Increased SST variability is observed in parts of the western and central Mediterranean Sea, while decreased variability is found in most of the eastern basin, alongside the underlying basin-wide mean warming. Results potentially suggest a stronger climate change effect in the southern Aegean and most of the Levantine Seas where the highest warming rates are observed despite the reduced variability of SST over the study period.

Applying the trend attributional ratio (TAR) proposed by Marin et al. (2021) for the Mediterranean Sea, we find a basin-wide dominance of the long-term SST change in causing more extreme events, longer heat exposure and a greater accumulation of heat stress on an annual basis. Interannual variability mainly tends to mitigate the observed increase in these MHW properties, particularly in the eastern sub-basin. In contrast, interannual variability emerges as the primary driver of mean MHW intensity trends in western and central Mediterranean areas.

These findings are in agreement with Marin et al. (2021) who demonstrated that trends in mean MHW intensity are predominantly forced by interannual variability at coastal sites globally and in the western and central Mediterranean Sea in particular, based on multiple SST datasets covering a shorter period. In the eastern Mediterranean areas where the highest warming rates are observed, their findings for mean intensity trends at coastal locations show that mean warming dominates interannual variability, in line with our results. Moreover, they showed that trends of the yearly cumulative intensity and the number of MHW days are primarily attributed to the mean warming, in agreement with our findings for these metrics across the basin. This inter-comparison enhances confidence in the current results derived from applying their methods to an observational SST dataset (Table 1) spanning the satellite period and covering the entire Mediterranean Sea.

The current study is also in line with other recent studies that follow different approaches to investigate origins of MHW trends in the Mediterranean Sea. Our findings for the mean MHW intensity trends are consistent with Simon et al. (2023), although they focus on summer MHW activity, which is an index combining the occurrence, intensity, duration and spatial extent of MHWs. In particular, they showed that interannual variability explains one third of the total trend of summer activity in the western basin and the Adriatic Sea while our results suggest that interannual variability predominates over the mean warming signal in most of the western and central sub-basins for mean intensity trends based on all events throughout the year.

In addition, Martinez et al. (2023) suggest that the intensification of MHW conditions in the Mediterranean Sea is primarily driven by the mean SST warming in the basin. Their analysis focuses on MHW duration, cumulative intensity, spatial extent, frequency and maximum intensity, with the latter being the only metric directly comparable to our study. A difference with respect to our results appears in the maximum intensity derived from detrended SST data: While Martinez et al. (2023) report an insignificant positive trend (basin-averaged), we detect a significant positive trend—more pronounced in the western basin. Notably, both studies agree on the dominant contribution of mean warming to the long-term trends of maximum intensity (and the rest of the metrics as well) though through different approaches. Martinez et al. (2023) base their conclusion on the insignificant long-term trends of basin-averaged metrics obtained from the detrended dataset, while our study relies on weighting the mean SST warming and interannual SST variability within the TAR framework. Specifically, TAR for maximum intensity confirms the dominant role of the mean SST warming, in line with Martinez et al. (2023), but also highlights non-negligible contributions from interannual SST variability, especially in the Alboran and Ligurian Seas.

Overall, this study reveals a dominant role of the mean SST warming in increasing the severity, total heat exposure and maximum intensity of MHWs in the Mediterranean Sea with a notably smaller effect in mean MHW intensity, which is largely influenced by interannual variability. The sensitivity of the employed attribution metric to different methodological choices for climatological baselines is higher for the mean MHW intensity compared to the rest of the examined metrics. This differentiation further suggests a more complex and likely less predictable behaviour of this property. In this respect, Marin et al. (2021) show that mean intensity trends exhibit significant sensitivity to different datasets, in contrast to other metrics. Also consistent with our findings, Oliver et al. (2019) show that trends in the annual number of MHW days and the maximum intensity are mainly forced by mean warming in the basin, noting a less clear origin of the trends for the latter. Nevertheless, their results cannot be directly inter-compared to ours, as they use a substantially different methodology based on a statistical climate model. Moreover, Schlegel et al. (2019) show that while linear trends significantly increase MHW duration, their effect on the maximum intensity can be either positive or negative. Their finding aligns with our results on the increased uncertainty associated with mean intensity; however, since it is based on the maximum intensity of averaged MHWs, it cannot be directly comparable to ours. Therefore, further investigation is needed to understand the reasons behind the distinct behavior of mean intensity—whether in terms of trends, trend attribution, or sensitivity to climatological baselines.

Finally, while the approach used to decompose SST signals in this study is commonly employed, there are limitations in interpreting the origin of the different components. To specifically assess the impact of anthropogenic influence, a future study for the Mediterranean Sea could employ an ensemble approach using climate models. This would offer a clearer understanding of the effects of climate change on extreme SST events relative to the role of natural variability, while also minimizing the impact of single-product characteristics on MHW metrics.

**Data availability**

Information on the product used in this paper is included in Table 1.

**Author contribution**

DD defined the research problem. DD conducted the analysis and wrote the manuscript, with contributions from GK, SD, and MH. All authors contributed to the interpretation of results.

**Competing interests**

The contact author has declared that none of the authors has any competing interests

**Financial support**

This work was partly funded by the Copernicus Med-MFC (LOT reference: 21002L5-COP-MFC MED-5500) within the framework of the Copernicus Marine Service, supporting Dimitra Denaxa. Sofia Darmaraki was supported by the Hellenic Foundation for Research and Innovation (H.F.R.I.) under the 3rd Call of the "Research Projects to Support Post-Doctoral Researchers" scheme (Project Number 07077, acronym TExMed).

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
