# Peer review of "Insights into sea surface temperature variability and the impact of long-term warming on marine heatwaves in the Mediterranean Sea"

_State of the Planet, 2024_

## Author Response (AR1)

**General comment by the authors**

We would like to thank both reviewers for the effort and care with which they assessed our submission. We have revised the manuscript following their excellent recommendations. Please, find below a point-by-point response to their comments, followed by a separate section listing the changes implemented in the manuscript. For ease of reference, the reviewer's comments have been numbered (C1, C2, etc) and are presented in blue font, while the authors' responses are presented in black font.

**Reply to Reviewer 1**

In general, I think the study is interesting and the analysis robust. I have only two major concerns, one related to the data set used and the other more related to the motivations and implications of this work for the communities who might be interested (in particular Marine Protected Areas and aquaculture sector).

We thank the reviewer for pointing out these two important aspects. Our specific responses are provided following the reviewer's comments below.

**C1. Motivation and Innovative aspects**

Which are the main innovative aspects of this work with respect to the Marin et al. (2021) which focused on the coastal analysis and used an ensemble of different SST products? In their case motivations of the work were to provide an informative framework for coastal management since they focus on the analysis of coastal MHWs. In the present work the analysis has been extended to the whole Mediterranean. Apart from the scientific interest, what are the possible implications of this analysis for marine biodiversity and ecosystem functioning across various time scales and for marine economic activities at the regional scale? It would be nice if the authors could provide more clear motivations for this study at regional level (all the Mediterranean versus the coastal region) and the implications that their conclusions might have on adaptation measures at relevant time scales.

Thank you for this valuable feedback. Our primary motivation lies in advancing scientific understanding of mean and extreme warming conditions and their drivers across the entire Mediterranean basin. In this context, expanding the analysis of Marin et al. (2021) beyond coastal locations provides a more holistic understanding of the basin. However, we recognize the need to emphasize broader motivations in the manuscript, especially concerning potential implications for marine biodiversity and ecosystem functioning.

Before applying the method of Marin et al. (2021) for attributing MHW trends, our study begins with an analysis of SST variability and extremes (99th percentile) across the Mediterranean Sea. This part aims to provide essential context on how long-term changes in extremes and variability are distributed spatially throughout the basin. This part sets the stage for the subsequent MHW analysis but also highlights areas of the basin where extreme warming is particularly pronounced and therefore might be disproportionately impacted.

For example, we find that the Adriatic, Aegean and northern Levantine Seas show the highest trends of both SST and its 99$^{th}$ percentile, suggesting higher vulnerability in terms of both accumulated warming and extreme SST occurrences. Such evidence is potentially useful for informing regional management strategies.

The Mediterranean Sea is a biodiversity hotspot and one of the most sensitive marine regions to climate change. By including the open sea, we account for areas that support key ecological processes which can be disrupted by MHWs. For example, pelagic species, critical to marine food webs, may be affected by MHWs, with potential repercussions for the fishery industry. A climate risk assessment by Hidalgo et al. (2022) finds the highest risks associated with impacts of ocean warming on fisheries resources (e.g., catch composition, distribution changes), highlighting the southeastern basin as the most impacted for both pelagic and demersal fisheries. Importantly, they find geographic differences in terms of drivers and impacts and recommend regionally tailored adaptation strategies.

Moreover, MHWs pose significant risks to aquaculture, which is a rapidly expanding industry in the Mediterranean Sea. Apart from fish mortality, MHWs affect aquaculture by facilitating the proliferation of pathogens and disease outbreaks, which can lead to unmarketable fish and substantial financial losses (Cascarano et al., 2021). Offshore aquaculture is increasingly being considered as an alternative to mitigate the effects of coastal warming, as it may help alleviate the impacts of extreme water temperatures (Mengual et al., 2021). In this context, a better understanding of MHWs is essential across both coastal and offshore areas.

Likewise, Marine Protected Areas (MPAs), which host vulnerable marine species, such as marine mammals and turtles (Chatzimentor et al., 2023), could benefit from a broader understanding of MHWs across the basin. Given the growing need for climate-based conservation strategies to protect marine life, it is important to enhance our understanding of extreme warming conditions at regional scale. Such insights could inform protective measures such as identifying spatial refugia and establishing new MPAs, strengthening the resilience of Mediterranean marine life to climate change (e.g., Zentner et al., 2023; Bates et al., 2019).

Considering the above, we will enrich the introduction of the revised manuscript making our motivations more explicit and addressing the ecological and socio-economic relevance of the study. Possible implications on marine ecosystems and marine economic activities will be included providing references.

Bates, A. E., Cooke, R. S. C., Duncan, M. I., Edgar, G. J., Bruno, J. F., Benedetti-Cecchi, L., Côté, I. M., Lefcheck, J. S., Costello, M. J., Barrett, N., Bird, T. J., Fenberg, P. B., and Stuart-Smith, R. D.: Climate resilience in marine protected areas and the 'Protection Paradox,' https://doi.org/10.1016/j.biocon.2019.05.005, 1 August 2019.

Cascarano, M. C., Stavrakidis-Zachou, O., Mladineo, I., Thompson, K. D., Papandroulakis, N., and Katharios, P.: Mediterranean aquaculture in a changing climate: Temperature effects on pathogens and diseases of three farmed fish species, https://doi.org/10.3390/pathogens10091205, 1 September 2021.Chatzimentor, A., Doxa, A., Katsanevakis, S., & Mazaris, A. D. (2023). Are Mediterranean marine threatened species at high risk by climate change? Global Change Biology, 29(7), 1809–1821. https://doi.org/10.1111/gcb.16577

Chatzimentor, A., Doxa, A., Katsanevakis, S., and Mazaris, A. D.: Are Mediterranean marine threatened species at high risk by climate change?, Glob. Chang. Biol., 29, 1809–1821, https://doi.org/10.1111/gcb.16577, 2023.

Hidalgo, M., El-Haweet, A. E., Tsikliras, A. C., Tirasin, E. M., Fortibuoni, T., Ronchi, F., Lauria, V., Ben Abdallah, O., Arneri, E., Ceriola, L., Milone, N., Lelli, S., Hernández, P., Bernal, M., and Vasconcellos, M.: Risks and adaptation options for the Mediterranean fisheries in the face of multiple climate change drivers and impacts, ICES J. Mar. Sci., 79, 2473–2488, https://doi.org/10.1093/icesjms/fsac185, 2022.Mengual, I. L., Sanchez-Jerez, P., and Ballester-Berman, J. D.: Offshore aquaculture as climate change adaptation in coastal areas: sea surface temperature trends in the Western Mediterranean Sea, Aquac. Environ. Interact., 13, 515–526, https://doi.org/10.3354/AEI00420, 2021.

Mengual, I. L., Sanchez-Jerez, P., and Ballester-Berman, J. D.: Offshore aquaculture as climate change adaptation in coastal areas: sea surface temperature trends in the Western Mediterranean Sea, Aquac. Environ. Interact., 13, 515–526, https://doi.org/10.3354/AEI00420, 2021.

Zentner, Y., Rovira, G., Margarit, N., Ortega, J., Casals, D., Medrano, A., Pagès-Escolà, M., Aspillaga, E., Capdevila, P., Figuerola-Ferrando, L., Riera, J. L., Hereu, B., Garrabou, J., and Linares, C.: Marine protected areas in a changing ocean: Adaptive management can mitigate the synergistic effects of local and climate change impacts, Biol. Conserv., 282, https://doi.org/10.1016/j.biocon.2023.110048, 2023.

**C2. Methodology**

Marin et al. 2021 found important differences between the SST products mainly for the MHW mean intensity, which was well correlated to SST variability, suggesting sensitivity of this metric to the specific SST data set. Indeed, one of their conclusion is that an ensemble approach should be adopted to minimize the impact of the choice of SST product on MHW metrics. Given that, according to your results, the interannual variability is the dominant driver only of the MHW mean intensity, how robust is your result since you use only one SST product which is also re-gridded from the original 0.05° resolution to a much coarser one? Can you comment on this?

Even if it is possible to find all the details of the SST product used here in the documentation that the authors refer to in Table 1, I think it would be useful for any reader to have a short summary of the main characteristics of the product given the possible sensitivity of the results to the specificities of the SST satellite product.

Thank you for your comment. In this study we chose to use the satellite–derived Mediterranean SST product from Copernicus Marine which has been widely used and validated. This reprocessed product is a gap-free, gridded product derived from a combination of satellite observations and in situ data, with high temporal and spatial resolution (0.05° daily data). The product is extensively validated against drifter buoy measurements as outlined in its Quality Information Document (Pisano et al., 2023), confirming its suitability for MHW studies.

The dataset was regridded to a coarser resolution to ensure efficient processing and analysis of results across the basin, given the large data volume and the multiple experiments conducted. Importantly, this step did not compromise the accuracy of the results or the ability to capture the spatial patterns of interest, as the key features and variability relevant to the objectives of our analysis are effectively represented.

We note that our results for coastal locations are highly consistent with those of Marin et al. (2021) for all the commonly used MHW metrics, despite the difference in products

and the study period as well. In addition, the sensitivity experiments we performed changing the climatological baselines for MHW detection further support the key role of interannual variability for mean intensity trends. The differentiation of mean intensity from the other metrics is consistently observed across the three sensitivity experiments suggesting that a less predictable bahavior should be expected for this metric, in agreement with the findings of Marin et al. 2021 based on four SST products. This consistency with their study despite the different datasets gives us further confidence in the robustness of the current findings. Nonetheless, we believe it is important to acknowledge in the revised manuscript the benefits of employing an ensemble approach. The Methods section will also be enhanced with additional characteristics of the SST product, following your suggestion.

**Minor comments:**

**C3.** Line 99.

*What do you mean exactly by "MHWs derived from both datasets are therefore relative to the initial state of the study period (1982)"?*

Thank you for your comment. Long-term SST trends were removed from the SST time series before calculating the daily SST climatology. As noted in L.97, the climatology constructed based on the detrended SST dataset is used for the detection of MHWs in both the non-detrended and detrended datasets, following the approach of Marin et al. (2021). This climatology does not include contributions from the long-term SST trend to the SST variance, it therefore represents the climatological state at the start of the study period (1982). In this sense, MHWs are identified with respect to these climatological conditions, in both datasets. We will add a brief clarification in the revised manuscript based on this explanation.

**C4.** Lines 156-157

*Our findings therefore indicate that, on top of the underlying mean warming, a large part of the western and central Mediterranean basin mainly experiences increased variability…*

Fig.1c is not showing this. Most of the western Med shows decreased variability and in most of the central Med the trend of STD is not significant. I suggest that all the conclusions related to this statement should be revised.

Thank you very much for noting this. Indeed, rather than a "large part" of the western and central basin, there are only distinct areas that show statistically significant increase in SST variability. We will revise this sentence as follows:

"Our findings therefore indicate that, on top of the underlying mean warming, few distinct areas, primarily in the western basin, experience increased SST variability, while most of the eastern basin and the region south of the Balearic Islands show reduced variability."

We note that there are no conclusions based on the content of this (non-revised) sentence later in the manuscript, so no further revisions to conclusions are needed.

**C5.** Fig.2 shows the temporal evolution of all the different MHWs metrics, but not the linear trend (see Legend).

Thank you for noting this. We will remove "*and linear trends*" from the legend, as these are presented in the Table instead.

**C6.** The legend of Fig.2 and Table 2 report twice the definition of the geographical areas.

Thank you for your comment. To improve clarity, we will remove the duplicate and show the geographical areas in the first panel of Fig. 1, making it easier for readers to refer to.

**C7.** In general, the quality of the figures could be improved. In particular, the use of the same color bar but for different ranges of values in Fig.3 does not help an immediate interpretation.

Thank you for noting this issue. We agree that varying ranges can make interpretation less immediate. However, all metrics except for mean intensity have comparable ranges of TAR values. For this reason, using a different range of values only for mean intensity allows for a better visualization of the spatial distribution of TAR across the basin. This approach was a compromise between consistency (using the same color palette among MHW metrics for visualising TAR), and interpretability. For the latter, the legend alerts readers to this color bar difference for mean intensity.

**Reply to Reviewer 2**

**C1. General comments**

*In this manuscript, the authors explore sea surface temperature data to analyze long-term changes in MHW characteristics, distinguishing the mean warming and SST variability impacts. The study is interesting but the authors should better clarify the motivations of the study, as well as the implications on marine ecosystems providing references. By comparing to the recent studies (cited by the authors), the novelty of the manuscript should be better highlighted.*

Thank you for this valuable feedback. Our primary motivation lies in advancing scientific understanding of mean and extreme warming conditions and their drivers across the entire Mediterranean basin. In this context, expanding the analysis of Marin et al. (2021) beyond coastal locations provides a more holistic understanding of the basin. However, we recognize the need to emphasize broader motivations in the manuscript, especially concerning potential implications for marine biodiversity and ecosystem functioning.

Before applying the method of Marin et al. (2021) for attributing MHW trends, our study begins with an analysis of SST variability and extremes (99th percentile) across the Mediterranean Sea. This part aims to provide essential context on how long-term changes in extremes and variability are distributed spatially throughout the basin. This part sets the stage for the subsequent MHW analysis but also highlights areas of the basin where extreme warming is particularly pronounced and therefore might be disproportionately impacted.

For example, we find that the Adriatic, Aegean and northern Levantine Seas show the highest trends of both SST and its 99$^{th}$ percentile, suggesting higher vulnerability in terms of both accumulated warming and extreme SST occurrences. Such evidence is potentially useful for informing regional management strategies.

The Mediterranean Sea is a biodiversity hotspot and one of the most sensitive marine regions to climate change. By including the open sea, we account for areas that support key ecological processes which can be disrupted by MHWs. For example, pelagic species, critical to marine food webs, may be affected by MHWs, with potential repercussions for the fishery industry. A climate risk assessment by Hidalgo et al. (2022) finds the highest risks associated with impacts of ocean warming on fisheries resources (e.g., catch composition, distribution changes), highlighting the southeastern basin as the most impacted for both pelagic and demersal fisheries. Importantly, they find geographic differences in terms of drivers and impacts and recommend regionally tailored adaptation strategies.

Moreover, MHWs pose significant risks to aquaculture, which is a rapidly expanding industry in the Mediterranean Sea. Apart from fish mortality, MHWs affect aquaculture by facilitating the proliferation of pathogens and disease outbreaks, which can lead to unmarketable fish and substantial financial losses (Cascarano et al., 2021). Offshore aquaculture is increasingly being considered as an alternative to mitigate the effects of coastal warming, as it may help alleviate the impacts of extreme water temperatures

(Mengual et al., 2021). In this context, a better understanding of MHWs is essential across both coastal and offshore areas.

Likewise, Marine Protected Areas (MPAs), which host vulnerable marine species, such as marine mammals and turtles (Chatzimentor et al., 2023), could benefit from a broader understanding of MHWs across the basin. Given the growing need for climate-based conservation strategies to protect marine life, it is important to enhance our understanding of extreme warming conditions at regional scale. Such insights could inform protective measures such as identifying spatial refugia and establishing new MPAs, strengthening the resilience of Mediterranean marine life to climate change (e.g., Zentner et al., 2023; Bates et al., 2019).

Considering the above, we will enrich the introduction of the revised manuscript making our motivations more explicit and addressing the ecological and socio-economic relevance of the study. Possible implications on marine ecosystems and marine economic activities will be included providing references.

Bates, A. E., Cooke, R. S. C., Duncan, M. I., Edgar, G. J., Bruno, J. F., Benedetti-Cecchi, L., Côté, I. M., Lefcheck, J. S., Costello, M. J., Barrett, N., Bird, T. J., Fenberg, P. B., and Stuart-Smith, R. D.: Climate resilience in marine protected areas and the 'Protection Paradox,' https://doi.org/10.1016/j.biocon.2019.05.005, 1 August 2019.

Cascarano, M. C., Stavrakidis-Zachou, O., Mladineo, I., Thompson, K. D., Papandroulakis, N., and Katharios, P.: Mediterranean aquaculture in a changing climate: Temperature effects on pathogens and diseases of three farmed fish species, https://doi.org/10.3390/pathogens10091205, 1 September 2021.Chatzimentor, A., Doxa, A., Katsanevakis, S., & Mazaris, A. D. (2023). Are Mediterranean marine threatened species at high risk by climate change? Global Change Biology, 29(7), 1809–1821. https://doi.org/10.1111/gcb.16577

Chatzimentor, A., Doxa, A., Katsanevakis, S., and Mazaris, A. D.: Are Mediterranean marine threatened species at high risk by climate change?, Glob. Chang. Biol., 29, 1809–1821, https://doi.org/10.1111/gcb.16577, 2023.

Hidalgo, M., El-Haweet, A. E., Tsikliras, A. C., Tirasin, E. M., Fortibuoni, T., Ronchi, F., Lauria, V., Ben Abdallah, O., Arneri, E., Ceriola, L., Milone, N., Lelli, S., Hernández, P., Bernal, M., and Vasconcellos, M.: Risks and adaptation options for the Mediterranean fisheries in the face of multiple climate change drivers and impacts, ICES J. Mar. Sci., 79, 2473–2488, https://doi.org/10.1093/icesjms/fsac185, 2022.Mengual, I. L., Sanchez-Jerez, P., and Ballester-Berman, J. D.: Offshore aquaculture as climate change adaptation in coastal areas: sea surface temperature trends in the Western Mediterranean Sea, Aquac. Environ. Interact., 13, 515–526, https://doi.org/10.3354/AEI00420, 2021.

Mengual, I. L., Sanchez-Jerez, P., and Ballester-Berman, J. D.: Offshore aquaculture as climate change adaptation in coastal areas: sea surface temperature trends in the Western Mediterranean Sea, Aquac. Environ. Interact., 13, 515–526, https://doi.org/10.3354/AEI00420, 2021.

Zentner, Y., Rovira, G., Margarit, N., Ortega, J., Casals, D., Medrano, A., Pagès-Escolà, M., Aspillaga, E., Capdevila, P., Figuerola-Ferrando, L., Riera, J. L., Hereu, B., Garrabou, J., and Linares, C.: Marine protected areas in a changing ocean: Adaptive management can mitigate the synergistic effects of local and climate change impacts, Biol. Conserv., 282, https://doi.org/10.1016/j.biocon.2023.110048, 2023.

**Specific comments**

**Introduction**

**C2.** L.28: reference in 2014 does not seem to me adequate when addressing the period 1982-2022, especially knowing the acceleration in global warming in the last decade and recent years. More recent publications: Martinez et al. (2023) 1982-2021, Juza et al. (2021)→ 1982-2020.

Thank you for your comment. We will update this part to better align with the period examined in this study, including the most recent publications you mention.

**C3.** l. 30: I would suggest "warm oceanic events"

Thank you for this suggestion, it will be included in the revised manuscript.

**C4.** l. 30 replace "Marine Heatwaves (MHW)" by "marine heatwaves (MHWs)"

Thank you for this suggestion, it will be included in the revised manuscript.

**2.1 SST observations**

**C5.** The authors "re-gridded" the datasets to coarser resolution, for computational reasons. I do not understand such an exercise and degradation of the datasets.

Thank you for raising this point. The dataset was regridded to a coarser resolution to ensure efficient processing and analysis of results across the basin, given the large data volume and the multiple experiments conducted. Importantly, this step did not compromise the accuracy of the results or the ability to capture the spatial patterns of interest, as the key features and variability relevant to the objectives of our analysis are effectively represented.

**C6.** The authors could provide a table or list of the derived datasets for clarity.

Thank you for your comment. Please, note that we use a single dataset in this study, so there is not a list of datasets to provide.

**C7.** Are these datasets available in open access? It would be a real added value of the paper.

Thank you for noting this. The used dataset is a gridded observational dataset (L4, reprocessed) for SST in the Med Sea, available (open access) through the Copernicus Marine Catalogue, as shown in Table 1. All information included in Table 1 for this Copernicus product (official product name (ID), source, links for associated documentation), follow the guidelines provided for product references in the context of this special issue (Copernicus Ocean State Report #9). Despite the details that a reader may find in the documentation of Table 1, we agree that some further information on this SST satellite dataset can be provided within the text. We will enrich this part in the revised manuscript.

**C8.** Maybe the authors could be interested by the paper from Amaya et al., 2023 Amaya 2023https://www.nature.com/articles/d41586-023-00924-2

Thank you for mentioning this valuable publication which highlights the importance of clearly articulating detection methodologies as regards the climatological baselines. We agree that this clarity is essential for understanding warm extremes and their impacts on marine life. Our choice for the climatological baseline used for MHW detection is

currently described in the manuscript in L. 97-101. Nevertheless, we will enrich the description of our approach and motivation for the employed climatology to improve clarity.

**2.2 MHW analysis**

**C9.** l.78 "for a 5-day period" → add "at least" or "or more"

Thank you for noting this, it will be added in the revised manuscript.

**C10.** l.100: the authors want to directly compare their results to Marin et al. (2021). They will obviously find differences when using different periods…

Thank you for this comment. We totally agree, and this is expected also due to the different dataset used in the current study. However, despite these differences, our results for the coastal locations agree with Marin et al. 2021 for all MHW metrics, which suggests that there is not large sensitivity to these methodological choices. Most importantly, the differentiation of mean intensity among the other metrics is observed in both studies. In addition, the experiments testing the sensitivity of results to climatological baselines further support that a less predictable behavior should be expected for mean intensity, in agreement with Marin et al. 2021.

**C11.** l.105: I do not feel comfortable with the reference period 1982-2002, since it does not respect the 30-year period recommendation (Hobday et al., 2016).

Thank you for pointing this out. We acknowledge that the reference period 1982–2002 does not follow the 30-year period recommendation by Hobday et al. (2016). However, the use of this shorter period (21 years) in EXP3 was intentional, serving as part of a sensitivity test to evaluate how TAR values are affected by changes in the reference period length. This experiment is not intended as an alternative climatological baseline but as a methodological sensitivity test. Additionally, we note that the differentiation of mean intensity from other metrics is consistently observed across the three experiments and aligns with the findings of Marin et al. (2021), who based their analysis on a 25-year period.

To address your concern, we will enhance the description of EXP3 in the revised manuscript to clarify that it aims to explore the impact of using a shorter reference period on the attribution of trends.

**C12.** l. 122-125: It could be great to provide some references. The choice of the metrics is driven by marine ecosystems implications. It should be clarified.

Thank you for this suggestion. Although our primary motivation is advancing scientific understanding of the drivers of MHW trends across the basin, we agree that linking the selected metrics to implications for marine ecosystems adds valuable context.

The yearly cumulative intensity ($CI_{yearly}$) not only integrates the effects of MHW duration and intensity but also serves as a measure of the long-term thermal stress induced by MHWs. Including the cumulative effect of multiple events may be relevant for several species in the Mediterranean basin, such as gorgonian populations which are severely impacted by recurrent MHWs in the basin (Orenes-Salazar et al., 2023) or fish species such as the gilthead seabream whose thermal tolerance can be affected by past

exposure to thermal stress (Kir, 2020). The limited acclimatization capacity of the latter and its relatively narrow temperature range challenge its survival where strong temperature variations occur. Areas with the highest event extremity (as represented by $SI_{max}$) may therefore pose severe risks to such species. In addition, wild marine species can often escape unfavorable aquatic conditions, whereas farmed species confined to aquaculture environments are more vulnerable to warm extremes (Beever et al. 2017). Such distinctions in behavioral flexibility are also important for the selection of appropriate metrics for assessing impacts on different marine populations and environments.

Following your suggestion, we will add a short discussion in the Methods subsection ("Selected MHW metrics") based on the above.

Beever, E. A., Hall, L. E., Varner, J., Loosen, A. E., Dunham, J. B., Gahl, M. K., Smith, F. A., and Lawler, J. J.: Behavioral flexibility as a mechanism for coping with climate change, Front. Ecol. Environ., 15, 299–308, https://doi.org/10.1002/fee.1502, 2017.

Kır, M.: Thermal tolerance and standard metabolic rate of juvenile gilthead seabream (Sparus aurata) acclimated to four temperatures, J. Therm. Biol., 93, 102739, https://doi.org/10.1016/j.jtherbio.2020.102739, 2020.

Orenes-Salazar, V., Navarro-Martínez, P. C., Ruíz, J. M., & García-Charton, J. A. (2023). Recurrent marine heatwaves threaten the resilience and viability of a key Mediterranean octocoral species. Aquatic Conservation: Marine and Freshwater Ecosystems, 33(11), 1161–1174. https://doi.org/10.1002/aqc.3997

**C13.** Could you precise which method is used and the level of significance?

Thank you for pointing this out. Trends were computed using linear regression, and their statistical significance was assessed using the Mann-Kendall test at the 95% confidence level. Confidence intervals for the trend estimates were calculated based on the standard error of the regression coefficients, using the Student's t-distribution ($\alpha=0.05$). We will include this information in the Methods section.

**Results**

**C14.** l. 189: "mean intensity seems to increase" → are they statistically increasing? or not?

Yes, this trend is statistically significant. We provide the entire sentence here for clarity: *"Mean intensity seems to increase basin-wide when the long-term warming trend of the Mediterranean Sea is removed (Fig. 2f), with statistically significant trends of 0.12, 0.08, and 0.05 degC.dec$^{-1}$ for the western, central and eastern sub-basins, respectively."* In addition, detailed information on the statistical significance of MHW trends for the basin and sub-regions is provided in Table 2.

**C15.** 222: "forced by interannual variability": what would be the results when distinguishing the seasons?

Thank you for raising this question. In this study, we use deseasonalized SST data in the entire analysis. Our aim is to focus on interannual variability and long-term trends without the effect of seasonal fluctuations, as described in Methods. Distinguishing the seasons could provide additional insights and is undoubtedly an interesting direction

for future research, it lies however outside the scope of the present work where seasonal variations are removed closely following the approach of Marin et al. (2021).

**C16.** l.229: "opposing trend of the (reduced) variability → decreasing trend of the variability

Thank you for noting this. It will be rephrased in the revised manuscript as suggested.

**Conclusions**

**C17.** l.284 "results potentially suggest"... A study over a more recent period could lead to different results. In particular, I think that the warming rate in WMED is higher than in EMED in the last decade... Have the two periods proposed in introduction/methodology been explored and analyzed to state such a conclusion?

Thank you for giving us the chance to comment on this. You are correct in noting that the warming rate in western is higher than in eastern basin over the most recent years. SST trends for the first and second half of the study period (1982-2002 and 2003-2023) reveal increased (decreased) warming rates over the western (eastern) basin for the second period, as shown in the figure below:

[Figure]

However, the conclusion in the quoted sentence is based on the entire study period (1982–2023), during which cumulative warming in the eastern basin is indeed larger than in the western basin as shown in Fig. 1 of the manuscript. Therefore, we believe that the statement remains accurate and not misleading within the context of the full study period. Using different time periods naturally results in varying trends, and we specifically chose to present the analysis over the longest available satellite record to ensure consistency and a comprehensive view of the trends. This was the focus of our study, as the main objective was to attribute MHW trends, rather than focusing on shorter-term trends.

For clarity, in the introduction/methodology we do not propose examining two periods. A different period than the entire 42-year period is used only within one of the

sensitivity tests (EXP3), in order to understand how the choice of the reference climatology period (i.e., its temporal coverage for EXP3) affects the trend attribution results.

**C18.** l.290: I would delta "in most of the basin, mainly"

Thank you for your comment. We agree, this will be corrected in the revised manuscript.

**C19.** l.293: I would delete the parenthesis

Thank you for your comment. We agree, this will be corrected in the revised manuscript.

**C20.** l.303-305: see my previous comments concerning the seasons.

Thank you for your comment. As explained in our answer to your relevant comment, this study aims to provide an annual perspective, in contrast to prior studies focusing on summer MHWs (e.g., results of Simon et al. 2022 for summer MHW Activity discussed in these lines).

**C21.** l.313-315: to be referred.

Thank you for your comment. We agree, this will be corrected in the revised manuscript.

**C22.** l.316-317: this statement has been repeated several times I think…

Thank you for your comment. You are right that there is a repetition of this statement. We believe however it is not redundant in the context of the conclusions section, as it constitutes the key finding of this work.

**Figures**

**C23.** Maybe boxes could be added in Figure 1.

Thank you for your suggestion. We agree that it is better to show the geographical areas in the first panel of Fig. 1, making it easier for readers to refer to. We will adjust Fig. 1a in the revised manuscript, as suggested.

**C24.** Figure 4 is not necessary

Thank you for your comment. We understand that key findings from the sensitivity experiments (EXP1-3) are described in L253-267. Nevertheless, we believe that Fig. 4 adds value by presenting these results in a more immediately interpretable format. The bar graphs, shown separately for each experiment, allow for clear visual comparisons both among the examined metrics within each experiment and across experiments. For this reason, we would prefer to keep this figure in the manuscript.

**Changes to the manuscript**

We have revised the manuscript, according to the suggestions and comments of the reviewers. All modifications are visible in the manuscript version that includes tracked changes. Please, note that references to lines are based on the revised document's line numbering. Reviewer comments have been numbered (C1, C2, etc) for ease of reference.

**Abstract:**

Line 16: Correction following C4 of Reviewer 1.

Lines 10, 23-24: Slight revisions to improve readability.

**Introduction:**

Line 28-29: The range of trend values is updated following the inclusion of additional references, as suggested in C2 of Reviewer 2.

Line 30: Marine Heatwaves → marine heatwaves, following C4 of Reviewer 2

Line 48-69: The introduction has been significantly enriched following both reviewers' request to better clarify our motivation and address the ecological and socio-economic relevance of the study. These additions strictly follow our answer to C1 of both Reviewers.

**Data and methods:**

Line 75-77: Additional characteristics of the used product are added following C2 of Reviewer 1.

Line 84-86: Information on confidence intervals and statistical significance is added following the C13 of Reviewer 2.

Line 119-120: A more detailed explanation for the selected climatology is added following the relevant clarification asked in C3 of Reviewer 1.

Lines 126-27: Clarification is added in the text following the concern of Reviewer 2 in C11.

Lines 133-34 & 147-154: The text is enriched with information on the potential usefulness of $CI_{yearly}$ and $SI_{max}$ including references, as suggested in C12 by Reviewer 2.

**Results:**

Figure 1: The separation into the 3 sub-regions is now explained only once to avoid repetitions, as noted in C6 of Reviewer 1. Now this information is included in the legend of Fig. 1 and, following the C23 of Reviewer 2, boundaries are also shown in Fig. 1a through the red dashed lines, as follows:

[Figure]

Line 186-188: The sentence is revised based on C4 of Reviewer 1.

Figure 2: Legend is corrected following C5 of Reviewer 1

Figure 3: Minor change in figure formatting (distances between sub-figures).

Line 208: Slight revision following C4 of Reviewer 1.

Line 257: Change in wording following C16 of Reviewer 2

**Summary and conclusions:**

Line 310: Slight revision following C4 of Reviewer 1.

Line 318: Sentence is adjusted following C18 of Reviewer 2

Line 321: Parentheses deleted as suggested in C19 of Reviewer 2

Line 347: Slight revision to improve readability.

Line 351-353: This sentence is enriched. It now comments on the benefit of employing an ensemble approach in future studies to minimize the impact of single-product characteristics on MHW metrics, following the relevant comment of Reviewer 1.

Line 363-365: Addition of financial support

**References**

Corrections in formatting, and addition of the following references based on our answers to C1 of both Reviewers, and C2 and C12 of Reviewer 2:

Bates, A. E., Cooke, R. S. C., Duncan, M. I., Edgar, G. J., Bruno, J. F., Benedetti-Cecchi, L., Côté, I. M., Lefcheck, J. S., Costello, M. J., Barrett, N., Bird, T. J., Fenberg, P. B., and Stuart-Smith, R. D.: Climate resilience in marine protected areas and the 'Protection Paradox,' https://doi.org/10.1016/j.biocon.2019.05.005, 1 August 2019.

Beever, E. A., Hall, L. E., Varner, J., Loosen, A. E., Dunham, J. B., Gahl, M. K., Smith, F. A., and Lawler, J. J.: Behavioral flexibility as a mechanism for coping with climate change, Front. Ecol. Environ., 15, 299–308, https://doi.org/10.1002/fee.1502, 2017.

Cascarano, M. C., Stavrakidis-Zachou, O., Mladineo, I., Thompson, K. D., Papandroulakis, N., and Katharios, P.: Mediterranean aquaculture in a changing climate: Temperature effects on pathogens and diseases of three farmed fish species, https://doi.org/10.3390/pathogens10091205, 1 September 2021.

Chatzimentor, A., Doxa, A., Katsanevakis, S., and Mazaris, A. D.: Are Mediterranean marine threatened species at high risk by climate change?, Glob. Chang. Biol., 29, 1809–1821, https://doi.org/10.1111/gcb.16577, 2023.

Hidalgo, M., El-Haweet, A. E., Tsikliras, A. C., Tirasin, E. M., Fortibuoni, T., Ronchi, F., Lauria, V., Ben Abdallah, O., Arneri, E., Ceriola, L., Milone, N., Lelli, S., Hernández, P., Bernal, M., and Vasconcellos, M.: Risks and adaptation options for the Mediterranean fisheries in the face of multiple climate change drivers and impacts, ICES J. Mar. Sci., 79, 2473–2488, https://doi.org/10.1093/icesjms/fsac185, 2022.

Kır, M.: Thermal tolerance and standard metabolic rate of juvenile gilthead seabream (Sparus aurata) acclimated to four temperatures, J. Therm. Biol., 93, 102739, https://doi.org/10.1016/j.jtherbio.2020.102739, 2020.

Mengual, I. L., Sanchez-Jerez, P., and Ballester-Berman, J. D.: Offshore aquaculture as climate change adaptation in coastal areas: sea surface temperature trends in the Western Mediterranean Sea, Aquac. Environ. Interact., 13, 515–526, https://doi.org/10.3354/AEI00420, 2021.

Orenes-Salazar, V., Navarro-Martínez, P. C., Ruíz, J. M., and García-Charton, J. A.: Recurrent marine heatwaves threaten the resilience and viability of a key Mediterranean octocoral species, Aquat. Conserv. Mar. Freshw. Ecosyst., 33, 1161–1174, https://doi.org/10.1002/aqc.3997, 2023.

Juza, M. and Tintoré, J.: Multivariate Sub-Regional Ocean Indicators in the Mediterranean Sea: From Event Detection to Climate Change Estimations, Front. Mar. Sci., 8, https://doi.org/10.3389/fmars.2021.610589, 2021.

Zentner, Y., Rovira, G., Margarit, N., Ortega, J., Casals, D., Medrano, A., Pagès-Escolà, M., Aspillaga, E., Capdevila, P., Figuerola-Ferrando, L., Riera, J. L., Hereu, B., Garrabou, J., and Linares, C.: Marine protected areas in a changing ocean: Adaptive management can mitigate the synergistic effects of local and climate change impacts, Biol. Conserv., 282, https://doi.org/10.1016/j.biocon.2023.110048, 2023.

---

## Author Response (AR2)

**Reply to reviewer**

The revised version of the manuscript is improved but it still needs some work to refine the text with the final aim to convey clear messages.

We thank the reviewer for the time they put in our manuscript. Please, find below our responses to their remaining comments. For ease of reference, the reviewer's comments are presented in blue and the authors' responses are presented in black font.

Fig.1. It is very hard to see the two dashed red lines which defines the three regions. Please use a thicker line or any other way to make the boundaries more evident. Furthermore, I do not find in the text how the three regions have been chosen.

Thank you for this comment. Figure 1 has been updated including thicker lines for marking the boundaries. The separation into the three sub-basins shown in Fig1a is now introduced also in Methods Sect. 2.2.

Section 3.2, in particular the first part related to Fig.2, needs to be revised and focus on some key messages otherwise the text now sounds simply as a description of the figure. The few considerations are not clear, for example the comment on the maximum severity index (lines 213-217) should be explained better. But the most important thing is to revise the text in order for the readers to capture more easily the relevant results and understand which are the main differences in the three regions and for the different metrics and the following implications.

Thank you for this comment. We agree that the description of Fig. 2 and Table 2 should be revised for clarity. We have updated this part in the revised manuscript, where all changes with respect to the previous version are tracked. Below we provide the improved explanation of the "contrasting" behaviour of $I_{max}$ and $SI_{max}$ specifically mentioned by the reviewer:

The maximum intensity ($I_{max}$) shows a higher positive trend in the western (0.38 degC.dec$^{-1}$) than in the eastern sub-basin (0.32 degC.dec$^{-1}$) (Fig. 2b). In contrast, the maximum severity index ($SI_{max}$) exhibits a stronger increasing trend in the eastern (0.3 units.dec$^{-1}$) than in the western sub-basin (0.23 units.dec$^{-1}$) (Table 2). This pattern arises potentially due to $SI_{max}$ quantifying the extremity of MHW intensity relative to a fixed climatological threshold (Eq. 4). Since this threshold remains constant over time, the larger increase in $SI_{max}$ in the eastern sub-basin suggests that maximum intensities in this region are becoming proportionally more extreme compared to historical conditions. The observed differences in trends between $I_{max}$ and $SI_{max}$ across sub-basins highlight that absolute MHW intensities are increasing more rapidly in the western sub-basin, while their relative extremity compared to the historical baseline is increasing more in the eastern sub-basin.

Section 3.2. The results of EXP2 and EXP3 show opposite behavior with respect to EXP1 for the mean intensity TAR in terms of the percentage of the total number of

Mediterranean grid points (Fig.4). Can you suggest any hypotheses for this? Why do the two different climatologies have an opposite impact and which are the implications? You might want to see two recently relevant published papers:

-Smith, Kathryn E., et al. "Baseline matters: Challenges and implications of different marine heatwave baselines." Progress in Oceanography 231 (2025): 103404.

-Capotondi, A., Rodrigues, R. R., Sen Gupta, A., Benthuysen, J. A., Deser, C., Frölicher, T. L., ... & Wang, C. (2024). A global overview of marine heatwaves in a changing climate. Communications Earth & Environment, 5(1), 701.

Thank you for your thoughtful question.

The sensitivity tests aimed to assess whether the approach used to compute the climatology affects results on the origin of MHW trends. EXP1 follows the approach of Marin et al. (2021) using a detrended baseline for the computation of climatology in the non-detrended dataset, while EXP2 and EXP3 use fixed baselines with reference periods spanning 1982-2023 and 1982-2002, respectively.

TAR values indicate that mean intensity (Imean) is the only metric whose trends are significantly affected by interannual variability. In turn, these sensitivity tests reveal a further differentiation of Imean, this time in relation to the climatology used. The relative role of mean SST warming against the interannual variability decreases from EXP2 to EXP3 for Imean, while the origin of trends for all other metrics shows negligible sensitivity to the choice of climatology. In particular, EXP2 shows the greatest percentage of grid points with positive Imean TAR (46%) compared to EXP1 (34%), whereas EXP3 exhibits a lower percentage (30%).

As expected, EXP3 has a cooler baseline and threshold compared to EXP2, leading to generally higher MHW activity in EXP3. We note here that recent research shows less intense MHWs for shorter baseline periods (e.g., Richaud et al., 2024; Darmaraki et al., 2025), which, however, is due to the shorter period spanning more recent and thus warmer years. In our study, for all examined metrics, we find higher values and trends for EXP3 relative to EXP2 throughout the basin, except for Imean, which shows lower trends in EXP3 relative to EXP2 in the western basin.

TAR fields for Imean are shown below also for EXP2 and EXP3 (not included in the manuscript). While the spatial distribution of TAR is very similar to EXP1, slightly lower TAR values are observed in EXP3 compared to EXP2. As shown in Fig. 4, fewer positive-TAR grid points (i.e. where mean warming dominates interannual variability) are observed in EXP3 compared to EXP2, which, as illustrated in the TAR fields, is more pronounced in the southwestern basin.

**EXP2:**

[Figure]

**EXP3:**

[Figure]

Overall, the sensitivity tests confirm the key conclusions of the main experiment (EXP1) for all examined metrics and highlight the distinct behaviour of Imean both with respect to other metrics and to different approaches for climatology. Importantly, previous studies have also noted distinct characteristics of MHW intensity. Marin et al. (2021) have shown that Imean trends are highly affected by internal variability and exhibit significant sensitivity (in contrast to other metrics) to different datasets. In addition, Oliver et al. (2019) have found a less clear origin of the trends of MHW intensity at global scale, though based on the maximum MHW intensity. Moreover, Schlegel et al. (2019) have shown that while the effect of linear trends on MHW duration is a significant increase, the effect on the maximum intensity can be either positive or negative. Their finding aligns with our results on the increased uncertainty associated with mean intensity; however, since it is based on the maximum intensity of averaged MHWs, it cannot directly comparable to ours.

We believe that further study should address the cause behind Imean's exception by investigating its behavior under different methodological choices (especially datasets and baselines) following an ensemble approach to ensure robustness. The revised conclusions section now includes the additional studies mentioned above in relation to the discussed finding and notes that further investigation is needed to understand the reasons behind the distinct behavior of mean intensity—whether in terms of trends, trend attribution, or sensitivity to climatological baselines.

Darmaraki, S., Krokos, G., Genevier, L., Hoteit, I., and Raitsos, D. E.: Drivers of marine heatwaves in coral bleaching regions of the Red Sea, Commun. Earth Environ., 6, https://doi.org/10.1038/s43247-025-02096-5, 2025.

Richaud, B., Hu, X., Darmaraki, S., Fennel, K., Lu, Y., and Oliver, E. C. J.: Drivers of Marine Heatwaves in the Arctic Ocean, J. Geophys. Res. Ocean., 129, https://doi.org/10.1029/2023JC020324, 2024.

Schlegel, R. W., Oliver, E. C. J., Hobday, A. J., and Smit, A. J.: Detecting Marine Heatwaves With Sub-Optimal Data, Front. Mar. Sci., 6, 1–14, https://doi.org/10.3389/fmars.2019.00737, 2019.

Conclusions. Lines 346-353. I found the consideration regarding the difference between the results of this work on the results from Martinez et al. (2023) complicated to follow. Can you please try to clarify which is the main message here?

Thank you for your comment. We agree that this part should be better explained. We provide below the revision of the lines referring to the discussed difference between the two studies:

Revised text:

In addition, Martinez et al. (2023) suggest that the intensification of MHW conditions in the Mediterranean Sea is primarily driven by the mean SST warming in the basin. Their analysis focuses on MHW duration, cumulative intensity, spatial extent, frequency and maximum intensity, with the latter being the only metric directly comparable to our study. A difference with respect to our results appears in the maximum intensity derived from detrended SST data: While Martinez et al. (2023) report an insignificant positive trend (basin-averaged), we detect a significant positive trend—more pronounced in the western basin. Notably, both studies agree on the dominant contribution of mean warming to the long-term trends of maximum intensity (and the rest of the metrics as well) though through different approaches. Martinez et al. (2023) base their conclusion on the insignificant long-term trends of basin-averaged metrics obtained from the detrended dataset, while our study relies on weighting the mean SST warming and interannual SST variability within the TAR framework. Specifically, TAR for maximum intensity confirms the dominant role of the mean SST warming, in line with Martinez et al. (2023), but also highlights non-negligible contributions from interannual SST variability, particularly in the Alboran and Ligurian Seas.

**Changes to the manuscript**

We have revised the manuscript, according to the additional comments of the reviewer. All modifications are visible in the manuscript version that includes tracked changes. Please, note that references to lines are based on the revised document's line numbering.

Line 4: Typo correction

**Data and methods:**

Line 109: The separation into the three sub-basins shown in Fig1a is now introduced in Methods

**Results:**

Figure 1 has been updated including thicker lines for marking the boundaries

Lines: 204-243: Revision of results' description to improve explanations and highlight key findings.

**Summary and conclusions:**

Lines: 342-353: Revision of text to provide a cleared description of the comparison between the studies

Lines: 359-369: Updated discussion on the distinct behavior of the mean intensity.

**References**

Line 389: Addition of reference:

Schlegel, R. W., Oliver, E. C. J., Hobday, A. J., and Smit, A. J.: Detecting Marine Heatwaves With Sub-Optimal Data, Front. Mar. Sci., 6, 1–14, https://doi.org/10.3389/fmars.2019.00737, 2019.

**Financial support**

Line 384: Update of Financial support field